# KRAS-Mutant Lung Cancer: Targeting Molecular and Immunologic Pathways, Therapeutic Advantages and Restrictions

**DOI:** 10.3390/cells12050749

**Published:** 2023-02-26

**Authors:** Nastaran Karimi, Seyed Javad Moghaddam

**Affiliations:** 1Faculty of Medicine, Marmara University, Istanbul 34899, Turkey; 2Department of Pulmonary Medicine, The University of Texas MD Anderson Cancer Center, Houston, TX 77030, USA; 3The University of Texas MD Anderson Cancer Center UTHealth Houston Graduate School of Biomedical Sciences, Houston, TX 77030, USA

**Keywords:** lung cancer, KRAS, molecular pathways, therapy

## Abstract

*RAS* mutations are among the most common oncogenic mutations in human cancers. Among *RAS* mutations, KRAS has the highest frequency and is present in almost 30% of non-small-cell lung cancer (NSCLC) patients. Lung cancer is the number one cause of mortality among cancers as a consequence of outrageous aggressiveness and late diagnosis. High mortality rates have been the reason behind numerous investigations and clinical trials to discover proper therapeutic agents targeting KRAS. These approaches include the following: direct KRAS targeting; synthetic lethality partner inhibitors; targeting of KRAS membrane association and associated metabolic rewiring; autophagy inhibitors; downstream inhibitors; and immunotherapies and other immune-modalities such as modulating inflammatory signaling transcription factors (e.g., STAT3). The majority of these have unfortunately encountered limited therapeutic outcomes due to multiple restrictive mechanisms including the presence of co-mutations. In this review we plan to summarize the past and most recent therapies under investigation, along with their therapeutic success rate and potential restrictions. This will provide useful information to improve the design of novel agents for treatment of this deadly disease.

## 1. Introduction

Lung cancer is among the cancers with the highest mortality rate and an approximate 5-year survival rate of 20%. Around 80% of total lung cancer cases happen to be non-small-cell lung cancer (NSCLC) with adenocarcinoma as the most common histological form [1,2,3]. *RAS* mutations are the most common oncogenic mutations in human cancers and *KRAS* has the highest frequency among other members of the *RAS* family in NSCLC. Approximately 30% of NSCLC cases in western countries are *KRAS* mutated, while Asian patients with NSCLC are estimated to be 10% positive [4,5,6,7]. *KRAS*-mutant NCSLC frequently emerges with other common mutations including mutations of tumor suppressor genes via loss of function and deletions. *TP53, STK11, KEAP/NFE2L2* gene mutations have been shown to be associated with *KRAS*-mutant lung cancers, and controversial prognostic values of the co-mutations have been declared along with different therapeutic outcomes in multiple clinical trials [8,9,10]. Various therapies have been conducted for the purpose of stopping the *KRAS*-mutant NSCLC in its primary and advanced phases, however, none of the therapies including chemotherapies have shown optimistic results. In this review, we plan to summarize the past and most recent therapies under investigation along with their success rate and potential restrictions. This will provide useful information to improve the design of novel agents for the treatment of this deadly disease.

## 2. Biology of *KRAS*

### 2.1. Function and Downstream Effector Pathways

The most frequently evaluated forms of the human *RAS* gene family are Kirsten rat sarcoma viral oncogene homolog (*KRAS*), neuroblastoma rat sarcoma viral oncogene homolog (*NRAS*), and Harvey rat sarcoma viral oncogene homolog (*HRAS*) with 80%, 11%, and 3% contribution of all RAS isoform respectively [11]. *KRAS* encrypts a membrane-bound GTPase protein which transfers the mitogenic data from upstream receptors—such as the epidermal growth factor receptor (EGFR), fibroblast growth factor receptor (FGFR), and human epidermal growth factor receptors 2–4 (HER2–4/ERBB2–4)—to different intracellular downstream pathways including those which have the most impact on tumorigenesis, such as the mitogen activated protein kinase (MAPK) pathway (RAF/MEK/ERK) that regulates cell survival, proliferation, and differentiation [12]. The phosphatidylinositol-3-kinase (PI3K) pathway activates AKT leading to phosphorylation of several transcription factors including nuclear factor (NF)-κB, mammalian target of rapamycin (mTOR), and forkhead box O (FOXO), which in turn controls cell apoptosis, survival, metabolism, and cell-cycle [13]. Another pathway with an impact on cell survival and proliferation is the RAL guanine nucleotide dissociation stimulator (RALGDS-RA) [14]. These actions are consequences of an active RAS protein which is GTP-bound, and dephosphorylation of GTP to GDP inactivates this protein [12,15]. RAS activation is assisted via guanine nucleotide exchange factors (GEFs) through GTP-GDP substitution, whereas GTPase activating proteins (GAPs) aim to inactivate RAS through increasing the GTPase activity of RAS [16,17]. To participate in membrane-bound activity, the soluble RAS protein requires post-transitional modifications [18]. Spontaneous activation of the RAS protein (e.g., due to mutations) without upstream signaling leads to oncogenic cell proliferation and uncontrolled cell survival. Figure 1 illustrates the KRAS mechanism and downstream pathways.

### 2.2. KRAS Mutations in NSCLC and Associated Co-Mutations

*RAS* mutation frequently occurs at exons 2 and 3, with the highest ratio identified at codons 12 and 13 of the *KRAS* gene in NSCLC, which are mainly gain-of-function alterations fostering oncogenic manners [19,20]. Among these alterations, G12D (the substitution of Aspartic acid for Glycine 12) is the most common mutation in lung cancer patients with no history of cigarette smoking, while G12C (the substitution of Cysteine for Glycine 12) and G12V (the substitution of Valine for Glycine) are the two most frequently identified variations in smokers with lung cancer [21,22]. In an in vitro study, it was shown that each amino acid substitution has a different affinity to downstream KRAS pathways; G12C- and G12V-mutated *KRAS* cell lines have higher RAL activity downstream and lower AKT phosphorylation, whereas G12D mutated ones have higher PI3K-AKT activity comparing to other alterations and wild type [23]. The distinctive prognostic features of different *KRAS* subtypes have become a controversial matter, as one study supports the role of these different *KRAS* mutations as an overall survival prediction, while larger studies do not support these findings [23,24]. These oncogenic *KRAS* mutations and the co-occurrence of other alterations (*STK11* or *KEAP1*) encourage metabolic rewiring toward anabolism, furthering proliferation as well as redox management i.e., acquiring an adaptation to oxidative stress. In one of the latest studies on murine models, it was demonstrated that the lung cancer murine model with *KRAS* homozygosity had higher glycolytic activity which rewired the cell program towards increasing glutathione synthesis as an adaptation to antioxidation. Higher sensitivity to glycolytic inhibitor 2-deoxyglucose and the glutathione synthesis inhibitor buthionine sulfoximine, along with a lower survival rate, were reported among homozygote *KRAS*-mutant murine models compared to those with heterozygote alterations [8,25]. As mentioned above, *KRAS*-mutant lung cancer is commonly associated with co-mutations, which have been categorized into three expressing clusters in a recent cohort of early and advanced stage *KRAS*-mutant tumors via RNAseq analysis: co-mutations in serine/threonine kinase 11 (*STK11*) and tumor protein P53 (*TP53*), with worse overall survival as a co-mutation to *KRAS* in lung cancer; inactivation of cyclin-dependent kinase inhibitors 2A/B (*CDKN2A/B*), with enrichment in the expression of neoplastic gastrointestinal and wildtype P53 transcriptional activity; and low level of thyroid transcription factor-1 (TTF-1) expression [8]. The most frequent co-alteration associated with KRAS in 330 advanced NSCLC patients were *TP53* (41%)*, STK11* (28%)*, KEAP1/NFE2L2* (27%)*, RBM10* (16%)*,* and *PTPRD* (15%). In this study, the median overall survival (mOS) was 17 months for all KRAS-mutant patients. In the absence or presence of the *TP53* co-mutation, the reported OS was similar (P:0.5). The presence of *STK11* and *KEAP1* in patients was associated with a shorter OS (P:0.002 and *p* < 0.001 respectively) [10]. Next Generation Sequencing (NGS) analysis of 121 Brazilian patients with advanced NSCLC in a retrospective study demonstrated that 20.86% (24/115 valid samples) had *KRAS* mutations and 1 (0.87%) had *KRAS* amplification. G12D (6/24), G12V (5/24), G12C/G12A (3/24), and G13D/Q61L (2/24) were among the most frequent mutation varieties presented. Co-mutations associated with *KRAS* were noticed in 33.3% (8/24), of which 3/8 and 2/8 co-occurred with G12D and G12V alterations respectively [26]. A few studies suggest that the co-occurrence of *STK11* deletion is associated with worse overall survival prognosis, and that this co-mutation might characterize the therapeutic outcomes [8,27]. A study in China on 103 NSCLC patients reported that *KRAS* mutation was exclusively gender-related (*p* = 0.027) with high prevalence in male patients and having no association with age, smoking, and metastasis history. Moreover, this study mentioned that *KRAS* mutation occurrence in NSCLC patients in the Chinese population (5.8%) is similar to that of the same demographic of patients in East Asian countries and in the USA population, while this number was shown to be higher in Western countries. These findings assert an ethnicity dependence in the relevance of KRAS alteration in NSCLC patients [28]. Prior pulmonary disease history in patients, such as chronic obstructive pulmonary disease (COPD), was shown to be irrelevant to *KRAS* mutation in a study on 325 NSCLC patients, however there is insufficient evidence to support this claim [29]. A comprehensive genomic cohort study on 371 lung cancer patients revealed precise information regarding different genetic alterations and co-mutations in subtypes of lung cancer. *CDKN2A, KEAP1, STK11* were reported to be associated with smoking, while *CDKN2B* was considered to be age-related (*p* value: 0.038) [30].

## 3. Therapeutic Approaches for KRAS-Mutant NSCLC

### 3.1. Direct KRAS Targeting

Thanks to revolutionary technologies in pharmacological laboratories, and advanced perspectives into KRAS biology, scientists have started to develop strategies that could directly target KRAS oncoproteins. These recent studies have led to the identification of two groups of KRAS inhibitors, particularly by modifying cysteine 12 and targeting the effector binding switch II region of KRAS [31] or the nucleotides pocket binding site [32]. The nucleotide pocket beneath the switch II region of KRAS, which substitutes glycine with cysteine, renders a solvent near the active site, permitting a novel target for the direct inhibition of KRAS G12C [31,33,34]. Currently, there are two direct KRAS G12C inhibitors that have been advanced to clinical trials. Sotorasib (AMG 510), also known as LUMAKRAS^™^, developed by Amgen, is a small molecule that irreversibly inhibits cysteine 12 in the vicinity of pocket 2 of the switch II region, thus resulting in an inactive GDP-bound state. This molecule inhibits KRAS G12C by reducing the interaction between KRAS G12C and SOS and therefore blocking SOS activity in nucleotide exchange. This suggests that the KRAS G12C inhibitor lowers the affinity of this unit with nucleotide exchange factors and therefore traps KRAS G12C in a GDP-bound state. Assuming this to be a fact, it was inferred that cellular factors which affect the affinity of nucleotide exchange factors may modulate the inhibition of KRAS G12C [34,35,36]. A phase I/II randomized study evaluated the safety and efficacy of sotorasib (Code Break 100: NCT03600883) in 126 patients with advanced stage KRAS G12C mutated NSCLC who were pre-treated with platinum-based therapy and inhibitors of programmed death 1 (PD-1). Sotorasib was administered orally once per day at a dosage of 960 mg. The objective response rate (ORR) was 37.1% with a median response duration of 11.1 months. The median overall survival was 12.5 months with a disease control rate (DCR) of 80.6% in participants. Treatment-related adverse events (TRAEs) were reported in 69.8% of patients, with 19.8% grade 3 events. The majority of side effects related to the gastrointestinal system, such as diarrhea (31.7%), nausea (19%), and slightly elevated alanine aminotransferase (ALT) and aspartate aminotransferase (AST) (15.1%), offering low-grade hepatic toxicity. The U.S. Food and Drug Administration (FDA) recently granted sotorasib designation for the treatment of advanced *KRAS*-G12C-mutated NSCLC in adult patients [37]. Currently, there are two other trials investigating sotorasib with the purpose of identifying patients for whom sotorasib can be a beneficial first-line therapy (NCT04303780 and NCT04185883).

Adagrasib (MRTX849) is another direct KRAS G12C inhibitor that acts irreversibly and selectively. A phase I/II study called KRYSTAL-1 (NCT03785249), in patients with a similar profile to that mentioned for sotorasib, evaluated the ORR and DCR of adagrasib and the results showed that in 51 patients with NSCLC, with a dosage of 600 mg twice per day, the ORR and DCR were 45% and 96% respectively. TRAEs of grade 3 or 4 were reported in 30% of patients; those with the highest frequency were fatigue (6%) and increased AST/ALT (5%). In a subgroup of patients with the *STK-11* co-mutation the ORR was 64%, which may indicate the beneficial role of this co-mutation in therapeutic progress [38]. Needless to say, multiple clinical trials are investigating the impact of adagrasib as a monotherapy or in combination with other therapeutic agents, aspiring to a novel approach in the treatment of advanced NSCLC (NCT04613596, NCT04685135, NCT05472623, NCT05375994).

BI 1701963, a pan-KRAS SOS1 inhibitor, is the only SOS1 inhibitor that has entered a clinical trial. SOS1 is one of the GEFs mediators of RAS activation (all variations, including G12D/C/V), and suppressing the activity of this small protein leads to lower GTP-bound (i.e., active) KRAS levels, and further inhibition of MAPK pathway signaling. A phase I dose-escalating trial of BI 1701963 as a monotherapy or in combination with trametinib (a MEK inhibitor agent) in advanced or metastatic *KRAS*-mutant patients (NCT04111458), is currently active and investigating the maximum tolerated dose and safety of this agent; by 11 February 2020, it has reported three treated patients [39]. Another clinical trial is investigating the safety and tolerability of adagrasib and BI 1701963 as a combination therapy in advanced KRAS-mutant patients (NCT04975256). Table 1 summarizes other direct KRAS inhibitor clinical trials.

A toxin-mediated therapy has been developed using RAS/RAP1-specific endopeptidase (RRSP) from *Vibrio vulnificus*, which bisects RAS and RAP1 in the switch I region among Y32 and D33 residues. RRSP targets three isoforms of RAS and mutated RAS oncogenes at positions 12, 13, and 61 in both GTP-bound and GDP-bound forms, resulting in the termination of downstream signaling of ERK and subsequently a reduction in proliferation [40,41] In an in vivo study performed on three mouse xenograft models (WT or mutant RAS) scientists developed RRSP as a pan-RAS biologic inhibitor conjugated to diphtheria toxin. They observed a reduced tumor burden in *KRAS*-missense-alteration-carrying cell lines in colon cancer and NSCLC [41].

Table 1 lists other direct KRAS inhibitor clinical trials along with their required information.

### 3.2. Indirect KRAS Targeting

#### 3.2.1. EGFR Signaling

KRAS is activated as a consequence of the phosphorylation of EGFR, however targeting ERBB RTK signaling using erlotinib and gefitinib showed no impact on disease progression in *KRAS*-mutant lung cancers [42,43,44]. On the other hand, a recent study demonstrated that knocking out EGRF in a mouse model of *KRAS*-G12D-mutant lung cancer resulted in tumor regression [45], although compensation by a non-ERBB family was seen in the later stage of the disease. This could give insight into the notion that combinational therapies could be successful, and an example of this includes neratinib (a pan-ERBB inhibitor) in combination with a MEK inhibitor, which has been approved by the FDA for adjuvant treatment [46]. The prognostic value of ERBB inhibitors in *KRAS*-mutant lung cancer patients needs further evaluation to confirm the variability of these inhibitors.

#### 3.2.2. KRAS Membrane Association

RAS trafficking to the plasma membrane (PM) is a crucial step, as RAS family proteins can only be activated once in the PM. RAS proteins are globular hydrophilic proteins that gain their ability to attach to the membrane through a process called farnesylation. Adding farnesyl via farnesyl transferase is the rate-limiting step in the process. Once farnesylated, the proteins are transported to the endoplasmic reticulum (ER) membrane where CAAX processing takes place as an interaction with RAS-converting enzyme 1 (RCE1) and isoprenyl carboxyl methyltransferase (ICMT) enzymes. The KRAS protein is then transported to the PM via a process in which PDE6δ might be involved. Phosphorylation of the Serine 181 residue of the KRAS protein can lead to the detachment of the protein from the PM and its return to the ER membrane [47]. The first generation of drugs that inhibited farnesyl transferase, including tipifarnib and lonafarnib, did not show significant clinical improvement in NSCLC patients, which could be explained by the compensatory pathway of prenylation through geranylgeranyl transferase I (GGTase I) [48,49,50,51]. The evaluation of satirasib, a second-generation agent of farnesyl transferase inhibitors (FTIs), was ended due to lack of efficacy in *KRAS*-mutant lung adenocarcinoma patients [52]. In vitro and in vivo studies targeting other enzymes involved in the post-transitional modification of the RAS protein, such as RCE1 and ICMT, have been conducted and led to contrary results [53].

#### 3.2.3. Downstream Effectors

As mentioned above, different downstream pathways are activated via the activity of KRAS, leading to cell proliferation and survival (e.g., MAPK, PI3K, mTOR, FOXO, RALGDS-RA). Numerous clinical trials have investigated the effect of inhibition of these pathways in the progress of NSCLC patients [54].

RAF Inhibitors

Sorafenib was one of the first drugs used in attempts to target RAF/MEK/ERK pathways in *KRAS*-mutant NSCLC, and was shown to be ineffective [55,56]. In a mouse study, it was demonstrated that C-RAF, rather than BRAF, is responsible for oncogenic signaling in *KRAS*-mutant NSCLC [57]. Numerous studies are researching the possible effects of different kinase inhibitors in single or combination therapies (NCT04620330), to assess the efficacy of VS-6766 (a dual RAF/MEK inhibitor) as a monotherapy or in combination with defactinib in NSCLC patients with recurrency [3,58]. Other clinical trials are recruiting to evaluate the feasibility of VS-6766 in combination with sotorasib (NCT05074810) and adagrasib (NCT05375994) in *KRAS*-G12C-mutant NSCLC subjects.

MEK Inhibitors

MEK inhibitors that target other intermediates in KRAS/RAF/MEK/ERK pathways were abrogated in NSCLC tumor ablation as a single agent therapy due to ERK signaling activation. Nevertheless, combinational therapies utilizing MEK inhibitors with other agents showed some evidence of efficacy [59]. However, adverse effects followed in a phase I study (NCT2450656) of afatinib and selumetinib in *KRAS*-mutant subjects, with insufficient anticancer impact leading to no further trials for this combination [60]. Other combinational trials of MEK inhibitors are underway to further assess their safety and efficacy (NCT03581487, NCT03170206, NCT04967079).

PI3K, AKT, mTOR Inhibitors

PI3K/mTOR/AKT inhibitors are another group of cancer therapeutic agents that are commonly being tested in trials in combination with MEK inhibitors in the BTTLE-2 program (NCT01248247), which showed a better 8-week disease control rate compared to erlotinib. However, further investigation of better treatment strategies was suggested [61]. In another study, in vitro and in vivo studies of the second-generation KRAS-G12C-inhibitor ARS1620 in combination with PI3K inhibitors resulted in tumor regression in *KRAS*-mutant NSCLC [62]. A phase I study carried out to test pictilisib, a pan-PI3K inhibitor, in combination with first-line treatment options for NSCLC, demonstrated safety and promising anticancer abilities [63].

FAK Inhibitors

Focal adhesion kinase (FAK) is a non-receptor tyrosine kinase with a crucial role in cancer cell proliferation, migration, and adhesion to the extracellular matrix [64]. An in vivo study showed that RHOA-FAK signaling could be a potential target in *KRAS*-mutant NSCLC with INK4a/ARF and TP53 deficiency [65]. Based on these findings, a phase II trial (NCT01951690) assessed the efficacy and safety of defactinib, a FAK inhibitor, in previously treated NSCLC patients, although the clinical outcome was moderate with no relation to INK4a/ARF and *TP53* status [66].

HSP90 Inhibitors

Heat shock protein 90 (HSP90) is a chaperone with a particular N-terminus ATP binding domain that assists other protein folding, stabilizing the conformational state, and helping them adjust to heat. HSP90 inhibition was shown to disrupt the protein function and therefore cancer cell growth [67]. Ganetespib (STA-9090) is a potent HSP90 inhibitor that was evaluated in a randomized phase III (GALAXY II) study with docetaxel in previously treated NSCLC patients. The trial was terminated due to the inefficacy of the gantespib and docetaxel combination [68]. Luminespib (AUY922) is another drug of this group which demonstrated efficacy and disease regression as a monotherapy in previously treated NSCLC with *EGFR* gene mutations and *ALK* rearrangement, and had no particular impact on *KRAS*-mutant tumors (NCT01124864) [69]. SNX-5422, a novel HSP90 inhibitor, was evaluated in a phase I study with carboplatin and paclitaxel (an anti-microtubule agent) with a dose maintenance of SNX-5422 in NSCLC subjects. The study demonstrated the agent’s anti-cancer activity and good tolerance [70].

### 3.3. Targeted Metabolic Rewiring

#### 3.3.1. Metabolic Rewiring

The metabolic rewiring of tumor cells in order to compensate for the increased demand for energy as a result of tremendous tumor cell proliferation, particularly altered glucose metabolism, has been a topic of interest in cancer therapy studies [71]. In a study on fresh surgically removed NSCLC tumor specimens, an elevated level of lactate was illustrated, pointing toward increased glycolysis in tumor cells as well as increased levels of citrate and succinate, elements of the TCA cycle [72]. In various mutant *KRAS* model studies in vitro and in vivo, besides altered glucose metabolism, other metabolic reprogramming such as increased autophagy and macro-pinocytosis were shown to play a crucial role in the metabolic rewiring process [73,74]. The studies targeting metabolic rewiring in NSCLC in murine models have mainly been performed in three areas: increased amino acid utilization; lipid biosynthesis and beta oxidation (e.g., autophagy inhibitor); and glucose metabolism. Drugs that interfere with glucose metabolism such as metformin [75], an antihyperglycemic agent targeting mitochondrial glycerophosphate dehydrogenase, have been investigated in a different clinical trial to evaluate their possible therapeutic effect on NSCLC patients [76].

#### 3.3.2. Autophagy Inhibitors

Autophagy is a mechanism that contributes to the dynamic recycling of intracellular compartments in order to help cell homeostasis. There are three main autophagy methods: macro autophagy, micro autophagy, and chaperone-mediated autophagy [77]. There are contradictory results concerning the role of autophagy in cancer, as it can work as a tumor suppressor or as a mechanism for tumor cell survival [78]. Macro autophagy is the mechanism that *KRAS*-mutant cells depend on to maintain cellular homeostasis [79,80]. ‘Autophagy addiction’ was the name used in an article that reported the dependency of *HRAS* and *KRAS* mutated cells on autophagy, as the absence of it leads to mitochondrial waste-product, and TCA cycle intermediates build up in cells which eventually leads to apoptosis [79]. Various clinical trials have been conducted investigating the validity of the therapeutic role of autophagy inhibitors. Hydroxychloroquine is a lysosomotropic agent for autophagy blockage which has been widely under investigation for its role in the treatment of cancer. Particularly, a new study is being conducted to investigate the role of hydroxyquinoline and binimetinib, a selective MEK inhibitor, in 29 patients with advanced *KRAS*-mutant NSCLC (NCT04735068). Further trials need to be run to reveal the possible tumor-cell-suppressing role of autophagy inhibitors.

### 3.4. Synthetic Lethality Partner Inhibitors

Synthetic lethality is defined as the co-occurrence of two genetic contents that exploit cellular death. The main events which contribute to synthetic lethality include loss-of-function mutations, the over-expression of genes, and environmental factors [81]. This concept has drawn the attention of cancer biologists as a potential targeted therapy since various phenomena, including proteotoxic stress, oxidative stress, DNA damage, and metabolic stress, follow cancer cell metabolism. These conditions have manifested novel therapeutic avenues for cancer therapy [82]. KRAS signaling harbors cellular conditions known as non-oncogenic addiction, which is the notion behind the KRAS-signaling synthetic therapies [82]. RNA interference (RNAi) and CRISPR/Cas9 genetic screening methods have been applied to determine the genetics contributing to synthetic lethality in KRAS. However, these findings face various limitations in their application to the human cancer cell [83]. Reactive oxygen species (ROS) and hyper-replication of DNA lead to genomic toxicity in RAS signaling. Cancer cells compensate for these phenomena through the activation of DNA damage repair (DDR) checkpoints and pathways. It was found that wild type (WT) HRAS and NRAS have a crucial role in the activation of ATR/CHK1-mediated DDR, which leads to genomic instability in *KRAS*-mutant lung cancer cells. WT H/NRAS inhibit MAPK and AKT signaling, which in turn maintains the inhibition of CHK1; therefore, loss of function leads to impaired CHK1 activation and disturbed checkpoints and increases genomic instability and susceptibility to chemotherapies. Inhibition of the CHK1 pathway can be a suitable target for synthetic lethality in patients under chemotherapy [84]. Another study found that inhibition of nuclear trafficking via XPO1 leads to a synthetic lethal effect on *KRAS*-mutant cancer cells through the inhibition of transcription factor NF-κB, introducing a targetable notion for NSCLC therapies [85]. *KRAS*-mutant cancer cell metabolism is followed by endoplasmic reticulum stress and proteotoxic activation of unfolded protein response (UPR) pathways, which result in cell death. Scientists found that inhibition of HSP90 with the IPI-504 adjuvant, together with mTOR inhibition via Rapamycin, results in cell death and tumor regression in murine models [86]. A new study found that inhibition of NOP56, a ribosomal protein that plays a role in ribosomal assembly, leads to impairment in the response to ROS, therefore *KRAS*-mutant cells shifted their metabolism toward the mTOR pathway, and co-inhibition of NOP56 and mTOR led to tumor regression in xenograft mouse models [87,88]. Various clinical trials have been conducted to evaluate the feasibility of different therapeutic agents as synthetic lethality partner inhibitors; the proteosome inhibitor bortezomib, which was the first proteasome inhibitor approved by the FDA for mantle cell myeloma patients, is being clinically tested for NSCLC [89]. In a phase II (NCT00346645) study, the effectiveness of bortezomib on NSCLC patients with no prior chemotherapy was evaluated, but was terminated due to lack of efficacy. CDK4/6 inhibitors are another group of drugs including abemaciclib, ralimetinib, and palbociclib, whose efficacy in NSCLC disease progression is currently being investigated, whether as monotherapies or in combination with chemotherapies [90]

### 3.5. Immunotherapy

The over expression of co-inhibitory molecules of the immune system (e.g., CTLA4, PD-1) in tumor infiltrating lymphocytes (TILs), which leads to the suppression of T cell activity against cancer cells, has led to several studies investigating the immune checkpoint inhibitors (ICIs) as a potential agent in cessation of disease progression in multiple cancers, including NSCLC [91,92,93]. T cells are the immune system’s main weapon in the defense against cancer, and are regulated in response to antigen presenting cells (APCs) via major histocompatibility complex (MHC) [94]. There are co-inhibitory receptors that downregulate T cell response, functioning as ICIs, of which CTLA4 and PD-1 are the main ones [94]. CTLA4 exerts its inhibitory role by interacting with CD80 and CD86 on APCs while competing with the co-stimulatory receptor, CD28 [95]. High levels of CTLA4 and PD-1 expressions on TILs were reported in a study comparing NSCLC and healthy individuals [91], therefore monoclonal antibodies targeting these receptors are the main subject in immunotherapy against NSCLC.

#### 3.5.1. PD-1 Inhibitors

Pembrolizumab is a selective, fully humanized immunoglobulin (Ig) G4-kappa monoclonal antibody (mAb) targeting PD-1, which was initially approved by the FDA in 2011 for the treatment of melanoma [96]. In 2016, pembrolizumab was approved as a first-line therapy in NSCLC patients with PD-L1 >1% with no *EGFR* or *ALK* mutations [97]. Furthermore, in a phase III KEYNOTE-024 study (NCT02142738) pembrolizumab treatment in NSCLC patients with PD-L1 tumor proportion score of 50% or higher demonstrated higher progression-free survival compared to those treated with platinum-based chemotherapy [98]. In a randomized, placebo-controlled KEYNOTE-407 study (NCT02775435), pembrolizumab used as a first-line therapy in combination with carboplatin and paclitaxel/nab-paclitaxel improved the OS and progression-free survival [99].

Atezolizumab is another humanized Ig G4-kappa mAb interacting with PD-L1 which [100] in a randomized study, IMpower130 (NCT02367781), of previously untreated NSCLC patients with no *EGFR* or *ALK* mutations, demonstrated increased OS and progression-free survival in combination with chemotherapy compared to chemotherapy alone [101]. IMpower 110 (NCT02409342), a randomized phase III clinical trial on patients with stage IV NSCLC, which studied atezolizumab as monotherapy or in combination with chemotherapies that were selected based on tumor histology, showed an OS of 20.2 months compared to 13.1 months with chemotherapy as the only agent. IMpower 150 (NCT02366143) is another phase III trial that investigated chemotherapy with atezolizumab, chemotherapy with bevacizumab, and chemotherapy with both atezolizumab and bevacizumab in NSCLC patients as first-line therapy. This study indicated a higher OS in the chemotherapy plus atezolizumab arm compared to the other two [102].

Nivolumab is another PD-1 targeting mAb which has been investigated in combination with ipilimumab, a CTLA4 targeting agent, in two different studies called CHECKMATE-227 (NCT02477826) and CHECKMATE 9LA (NCT03215706). The results of CHECKMATE 9LA showed 15.6 months OS in previously untreated NSCLC patients treated with nivolumab and ipilimumab plus chemotherapy, compared to 10.9 months in patients who only received chemotherapy [102].

Cemiplimab, a PD-1 mAb, was recently approved by the FDA as studied in EMPOWER-Lung 3 trial (NCT03409614). It was a randomized double blind phase III study on NSCLC patients without *EGFR*, *ALK*, and *ROS* mutations, comparing cemiplimab in combination with chemotherapy against chemotherapy alone; the OS was 21.9 and 13.8 months respectively, indicating a clinically significant difference [103].

Besides atezolizumab, durvalumab is another PD-L1 mAb which was evaluated in a phase III PACIFIC study (NCT02125461) on patients with unresectable NSCLC without progression after chemotherapy. Recently, a three-year report of all patients’ OS has shown 57% OS in those receiving durvalumab versus 43.5% in the placebo group, which indicates a remarkable improvement in the treated group [104].

#### 3.5.2. CTLA-4 Inhibitors

Ipilimumab is an IgG1 mAb against CTLA4, and it has been approved by the FDA as a first-line therapy in combination with nivolumab in two different studies as mentioned above [102].

Tremelimumab is another CTLA4 mAb recently approved by the FDA in combination with durvalumab and platinum-based chemotherapy in metastatic NSCLC without *EGRF* and *ALK* gene mutations. The efficacy of this regimen was evaluated in a phase III, randomized, open label (NCT03164616) POSEIDON study with three treatment arms: (1) durvalumab + tremelimumab combination therapy + chemotherapy; (2) durvalumab monotherapy + chemotherapy; and (3) chemotherapy alone. The OS of the first arm was 14 months compared to 11.7 months in third arm, which signifies a clinical improvement in tremelimumab combinational therapy [105].

### 3.6. Other Immunomodalties

Tumor-promoting inflammation is a new concept of immune modality intervention, as it was demonstrated that inflammation has a fundamental role in tumorigenesis and progression in *KRAS*-mutant NSCLC. Manipulation of the immune system by cancer cells leads to the remodeling of immune cells in the tumor microenvironment (TME) and renders them incompetent [71]. Multiple cytokines, inflammatory signaling pathways, and innate immune signals play a crucial role in reprogramming the TME [106].

#### 3.6.1. Cytokines

Interleukin 1β (IL-1β) via its two receptors, IL-1RI andIL-1RII could send up-regulatory and down-regulatory signals, respectively. Multiple immune cells such as macrophages, natural killer cells (NK), neutrophils, and T cells secret IL-1β [106,107]. We found that blocking IL-1β with an IL-1β mAb in a *KRAS*-G12D-mutant mouse model of lung cancer led to tumor regression, which was associated with increased infiltration of CD8+ T cells, potentially due to a decrease in myeloid-cell-associated immunosuppression as well as inhibition of the NF-kB and STAT3 pathways [108]. Our findings were consistent with two other studies stating that an increased level of IL-1β is associated with NSCLC cell proliferation [109,110] and there is a negative correlation between IL-1β and progression-free survival of patients with NSCLC [109]. A canakinumab (anti-IL-1β) agent trial on atherosclerosis patients revealed decreased incidence of lung cancer in these patients compared to the placebo group [111]. Other trials are investigating the effect of canakinumab as a monotherapy (NCT03447769) or in combination with PD-1 mAbs such as pembrolizumab (NCT03968419) and durvalumab (NCT04905316) and chemotherapeutic agent, docetaxel (NCT03626545).

Interleukin 6 (IL-6) transfers signals to the cell via multiple pathways including RAF/MEK/ERK, mTOR, AKT/PI3K, and particularly JAK/STAT which fundamentally contributes to cell proliferation and survival [112]. IL-6 assists tumor cells’ survival with TME manipulation as well as downregulation of CD8+ T cells, macrophage M1 to M2 phenotype changes and upregulation of immune-suppressing cells such as T helper 17 (Th17), Tregs, and myeloid-derived suppressive cells (MDSCs) [106]. Advanced stage NSCLC patients with poor prognoses were found to have high levels of IL-6 [113,114,115]. The inhibition of IL-6 in *KRAS*-mutant lung cancer mouse models revealed lower pro-tumor characteristics in the TME, leading to tumor suppression [116,117]. Currently, three clinical trials are evaluating the efficacy of tocilizumab, an anti-IL-6 mAb, in combination with atezolizumab in locally advanced or metastatic NSCLC (NCT04691817), in combination with ipilimumab and nivolumab in melanoma, NSCLC, and urothelial cancer (NCT04940299), and in a multi-immunotherapy two-phased therapy in combination with chemotherapy or PD-1 inhibitors in metastatic NSCLC subjects (NCT03337698).

#### 3.6.2. Innate Immune Signaling

The innate immune signaling pathways are intermittent pathways operating as recognition strategies for the immune system, including stimulator of interferon gene (STING) signaling pathways, toll-like receptor pathways, and inflammasome. Targeting cells intrinsic STING signaling pathway is an emerging topic. STING is a fundamental element of the innate immune system that is located on the ER; it recognizes cytosolic DNA and further induces the synthesis of type I interferons (IFNs). Pathogenic dsDNA interacts with cyclic guanosine monophosphate-adenosine monophosphate synthase (cGAS) and further stimulates the production of 2′,3′-cyclic GMP-AMP (2′,3′-cGAMP). The interaction of 2′,3′-cGAMP and STING leads to the activation of TANK-binding kinase 1 (TBK1) and phosphorylation of interferon regulatory factor 3 (IRF3). IRF3 then moves to the nucleus and aids in the expression of IFN type I. Upregulation of the immune system via IFN type I is a well-known phenomenon, therefore STING agonist agents could be a potential novel therapy in cancer via the immune regulatory role of this pathway [118]. Currently, various clinical trials are evaluating the role of STING agonists in various cancer types, particularly solid tumors. An open-label, dose escalating phase I study (NCT04879849) of TAK-676 (STING agonist) and pembrolizumab after radiation therapy in different cancer patients, including NSCLCs, is currently recruiting. This could be a promising method which needs further research to assess the therapeutic potential of these agents as an anti-cancer therapy.

#### 3.6.3. Inflammatory Signaling Transcription Factors

Most the downstream pathways activated by KRAS eventually trigger two well-known transcription factors: STAT3 and NF-κB. STAT3, a member of signal transducer and growth factors (STAT), is one of the main transcription factors that transmit the signals of several cytokines, particularly IL-6 in *KRAS*-mutant tumor cells. The activation of JAK1/JAK2 signaling via IL-6 leads to the mobilization of STAT3 toward the nucleus, resulting in several gene translations responsible for angiogenesis and metastasis [119]. As a result of the administration of ruxolitinib (a JAK1/2 selective kinase inhibitor) in mouse models, a decline in tumor cell proliferation rate was reported, showing a positive correlation between JAK1/2 and disease progression in *KRAS*-mutant NSCLC [120]. However, in our recent in vivo study, we demonstrated a gender-specific correlation between STAT3 expression and *KRAS*-mutant NSCLC; female mice with tumor-cell-specific STAT3 deletion showed decreased tumorigenesis while males had the opposite outcome [121]. Several clinical trials (e.g., NCT02983578) are emerging to assess a novel therapeutic agent, AZD9150 (danvatirsen), an antisense oligonucleotide targeting STAT3 in NSCLC subjects. The inhibition of STAT3 with AZD9150 in xenograft models of lung cancer patients was associated with increased anti-tumor activity [122]. Another STAT3 inhibitor, TTI-101(C188-9) was shown to decrease tumor volume and growth in the A549 xenograft model of lung cancer [123] and it has entered a phase I trial as an oral STAT3 inhibitor in patients with advanced cancers, including NSCLC (NCT03195699). In a phase I study (NCT01184807) of OPB-51602 (STAT3 inhibitor) in NSCLC patients, antitumor activity was detected [124].

Another transcription factor that is activated as a result of the downstream pathways of KRAS is nuclear factor-κB (NF-κB), which is kept in its inactive state in naïve cells via proteins called inhibitor of κB (IκB). Activation of this pathway by cytokines or other inflammatory stimuli results in the phosphorylation of IκB by IκB kinase and releases the transcription factor into the nucleus, which will then stimulate several phenomena such as cell proliferation, survival, and inflammation [125]. Clinical studies found a positive correlation between high levels of IκB and tumor staging and metastasis state of lung cancer patients [126]. NF-κB activation has been observed in the TME and tumors of *KRAS*-mutant lung cancer mouse models [127]. However, targeting NF-κB can potentially render the immune system undefended, as it has an essential regulatory role in the innate and humoral response. Figure 2 demonstrates various therapeutic targets and the drug interventions which are directed at them.

#### 3.6.4. Chimeric Antigen Receptors (CAR) T Cells Immunotherapies

CAR T cells are emerging novel therapeutic immunological agents which are being widely investigated in cancer therapy research, and have been particularly successful in hematological malignancies. CAR T cells are genetically manufactured from patients’ isolated T cells to detect and bind to antigens on cancer cells. They consist of an intracellular domain, a transmembrane domain, and an extracellular antigen recognition domain which is highly receptive to tumor-associated antigens (TAA). To validate a therapy, the most crucial step is to define the particular TAA to the extent that it is only found on cancer cells. TAA heterogenicity in solid tumors has been a burden in contrast to hematological malignancies. Other challenges of CAR T cell therapy in solid tumors, in particular lung cancers, include the immunosuppressive TME, cytokine release syndrome, and neurological toxicity [128]. Some of the current targets for CAR T cells therapy in lung cancer are as follows: EGFR, HER2, mesothelin (MSLN), MUC1, CEA, PD-1, and CD80/CD86 [129]. Table 2 provides a comprehensive and detailed list of clinical trials mentioned, including the type of agents, study group, status, and phase of the studies.

## 4. The Mechanisms of Acquired On-Target and Off-Target Resistance to KRAS Inhibitors

An issue that burdens the expansion of KRAS inhibitors is the resistances that ultimately emerge despite disease regression in response to therapy. These acquired resistance mechanisms, which are defined as disease progression after 12 weeks of disease stability or regression, consist of on-target and off-target means [130].

‘On-target mechanism’ refers to those alteration in KRAS which hardens agent binding, and ‘off-target’ refers to those biological alterations which indirectly affect KRAS inhibition. Several studies have aimed to investigate the phenomena underlying resistance to KRAS inhibitors. An in vitro study employing *KRAS*-G12C-mutant Ba/F3 cells investigated the secondary *KRAS* mutations which can harbor resistance to sotorasib and adagrasib using N-ethyl-N-nitrosourea as a mutagenesis screen. Of the 142 clones which evolved resistance, 124 demonstrated multiple secondary *KRAS* mutations including Y96D and Y96S. Y96 plays a key role in hydrogen bond formation between KRAS and both sotorasib and adagrasib. Thus, alteration of this residue results in resistance to these inhibitors [131]. Another study on 38 clinical trial patients, including 27 with NSCLC, described an acquired resistance to adagrasib treatment in 17 (45%) individuals (including 10 (26%) NSCLC patients) via next-generation sequencing (NGS) sampling tissues or circulating tumor DNAs. These alterations were divided into three different classifications including secondary *KRAS* mutations (i.e., Y96C) on the switch II region, the binding site of KRAS for inhibitors, along with activating *KRAS* mutations in trans G12C, G12V, and G12W (on-target mechanisms). The second classification is those with MET amplification leading to RAS-ERK pathway reactivation and eventually higher level of GTP-bound RAS (off-target mechanism). The last group were those who had histological transformations to squamous cell carcinoma. It was noted that 7 (41%) out of 17 NSCLC patients displayed more than one coexisting resistance mechanism [132]. Multiple (10) acquired resistance mutations to adagrasib were found in a *KRAS*-G12C-mutant patient, which led to RAS-MAPK reactivation and *Y96D* mutation [133]. Note that other genomic alterations such as MAPK pathway genes or other genomic material affecting KRAS downstream pathways could potentially lead to resistance to KRAS inhibitors [130].

## 5. Role of Co-Mutations in Therapeutic Outcomes

As mentioned above, co-mutations associated with *KRAS* have different therapeutic impacts on treatment outcomes. The mutation of *TP53* or *STK11* in an in vivo study of *KRAS*-mutant tumors resulted in a failed response to docetaxel [134]. Patients with recurrent or metastatic NSCLC who were initially treated with a platinum agent, pemetrexed, +/- bevacizumab were evaluated to assess their co-mutation status. A shorter duration of therapy was linked to the presence of the *KEAP/NFE2L2* co-alteration (P: 0.008). However, *TP53* or *STK11* co-occurrence did not lead to significant variance in the duration of platinum-based treatment [10]. *KRAS/STK11* mutated NSCLCs are assumed to be unresponsive to ICIs as a result of a lack of TILs and low PD-L1 expression [135]. Another *KRAS/STK11* mouse model study established tumor regression following T cell infiltration subsequent to IL-6 antibody administration, while the same models were irresponsive to PD-1 mAb [136]. The presence of the *KEAP1* co-mutation in KRAS-mutant NSCLC is considered to cause resistance as well as lower OS in response to ICIs [10]. In a study investigating the impact of six *SWI/SNF* genes, including *SMARCA4*, on immunotherapy (e.g., CTLA4 and PD-1 inhibitors) outcomes in *KRAS*-mutant NSCLC, the worst OS and prognosis was associated with *SMARCA4* mutants, indicating the unfortunate outcome of this gene alteration with KRAS [137]. Despite these findings, co-mutations do not always correspond to the worse clinical scenario, as *KRAS/TP53* NSCLC patients responded well to therapies and had improved OS as a result. *TP53* co-alterations in these cases demonstrated an increased immune response and regulation, along with expression of the PD-1 immune checkpoint [135]. These findings indicate the need for a more individualized mutation-based therapy which needs to be further addressed.

## 6. Discussion

According to the recent findings regarding the therapeutic impact of co-mutations on the prognosis of *KRAS*-mutant NSCLC patients, in particular *SMARCA4* and *STK11* being associated with worse prognosis and *TP53* being correlated with improved immune response and OS. Genetic evaluation of patients with a *KRAS*-mutant NSCLC diagnosis is a key approach in order to define the best possible therapy. Furthermore, immune modalities that target the TME, including cytokines (IL-1, IL-6) and the STING pathway, have a fundamental role in tumorigenesis and progression. Their combination with other therapies, such as the direct targeting of KRAS (particularly the newly-FDA-approved sotorasib), conventional chemotherapy, and immune checkpoint blockade, could be a potential regimen which requires further assessment. We foresee the necessity of further studies that examine the targeting both of tumor-intrinsic factors and proinflammatory and immunosuppressive TMEs to better cover the complexity of *KRAS*-mutant NSCLC and provide more personalized treatment based on the genetic assessment of each patient.

## Figures and Tables

**Figure 1 cells-12-00749-f001:**
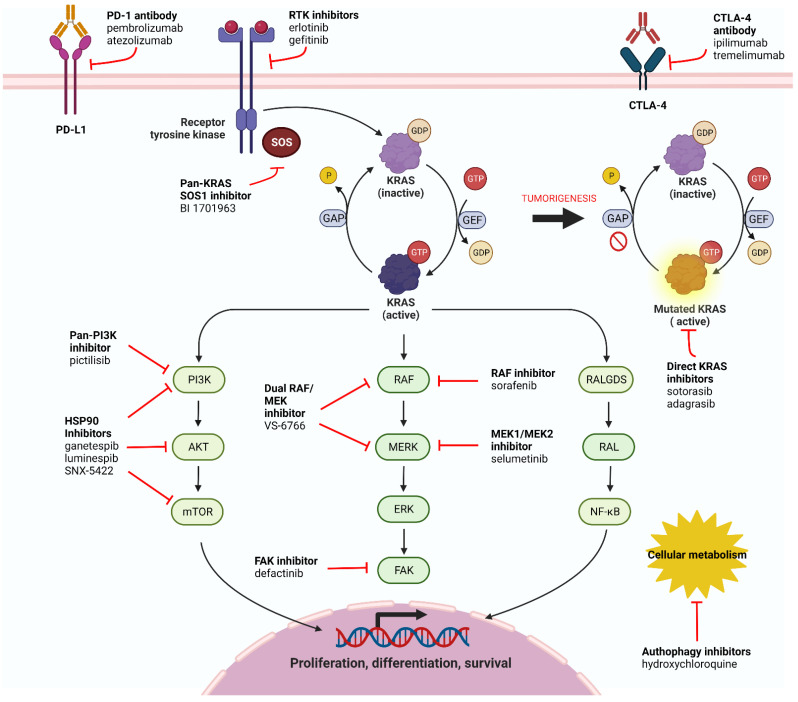
KRAS pathways with different target inhibitors. Activation of KRAS is shown in the upper part of the figure. Following the interaction with GEF, KRAS binds to GTP (active state) and GAP action leads to the release of phosphor, rendering KRAS inactive. On the right side, as a result of tumorigenesis GAP is shown inhibited leading to constant activation of KRAS. Following activation of KRAS multiple intracellular pathways become activated, resulting in cell proliferation, differentiation, and survival. Created with BioRender.com. Accessed on 21 February 2023.

**Figure 2 cells-12-00749-f002:**
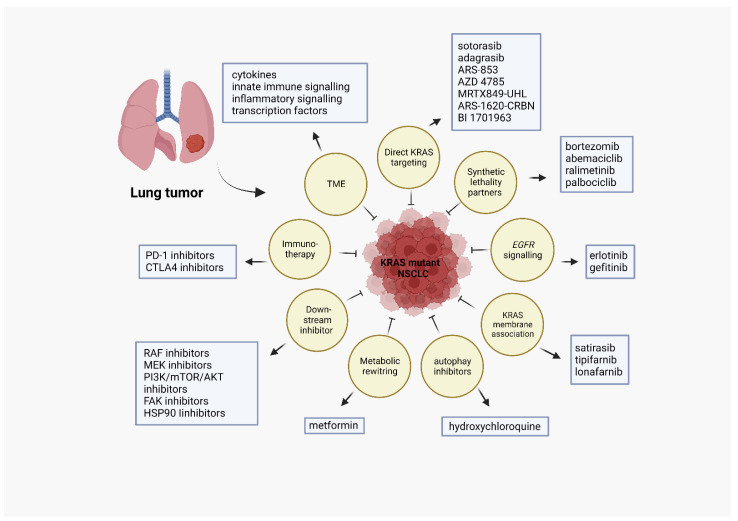
Different therapeutic targets in KRAS-mutant lung cancer with specific names of drugs approved or in trials along with tackled pathways. Created with BioRender.com. Accessed 17th February 2023.

**Table 1 cells-12-00749-t001:** This table includes other direct KRAS inhibitors clinical trials.

NCT Number.	Interventions	Title	Status	Conditions	Phases
NCT04585035	D-1553, Other	Study to Evaluate D-1553 in Subjects with Solid Tumors	Recruiting	Adult solid tumor, NSCLC, Colorectal Carcinoma	Phase I/II
NCT04956640	LY3537982, Abemaciclib, Erlotinib, Pembrolizumab, Temuterkib, LY3295668, Cetuximab, TNO155	Study of LY3537982 in Cancer Patients with a Specific Genetic Mutation (KRAS G12C)	Recruiting	Carcinoma, NSCLC, Colorectal Neoplams, Endometrial Neoplasms, Ovarian Neoplasms, Pancreatic Neoplasms	Phase I
NCT04449874	GDC-6036, Atezolizumab, Cetuximab, Bevacizumab, Erlotinib, GDC-1971, Inavolisib	A Study to Evaluate the Safety, Pharmacokinetics, and Activity of GDC-6036 Alone or in Combination in Participants with Advanced or Metastatic Solid Tumors with a KRAS G12C Mutation	Recruiting	NSCLC, Colorectal Cancer, Advanced Solid Tumors	Phase I
NCT04699188	JDQ443, TNO155, tislelizumab	Study of JDQ443 in Patients with Advanced Solid Tumors Harboring the KRAS G12C Mutation (KontRASt-01)	Recruiting	KRAS G12C Mutant Solid Tumors, NSCLC, Colorectal Carcinoma, Lung Cancer	Phase I/II
NCT05002270	JAB-21822, Cetuximab	JAB-21822 Activity in Adult Patients with Advanced Solid Tumors Harboring KRAS G12C Mutation	Recruiting	NSCLC, Advanced Solid Tumor, Colorectal Carcinoma	Phase I/II
NCT04006301	JNJ-74699157/ARS-3248	First-in-Human Study of JNJ-74699157 in Participants with Tumors Harboring the KRAS G12C Mutation	Completed	NSCLC, Advanced Solid Tumors, Colorectal Cancer	Phase I
NCT04165031	LY3499446, Abemaciclib, Cetuximab, Erlotinib, Docetaxel	A Study of LY3499446 in Participants with Advanced Solid Tumors with KRAS G12C Mutation	Terminated (due to unexpected toxicity findings)	NSCLC, Advanced Solid Tumor, Colorectal Carcinoma	Phase I/II

**Table 2 cells-12-00749-t002:** This table lists the clinical trials relating to indirect KRAS inhibitors, targeted metabolic rewiring agents, synthetic lethality partners, immunotherapies, and other immunomodalities.

NCT Number	Interventions	Title	Status	Conditions	Phases
NCT00346645	Bortezomib (Proteosome inhibitor)	A Phase II Study of Velcade^®^ in Patients Stage IIIB OR IV Non-Small Cell Lung Cancer	Terminated (stopped after planned interim analysis for lack of efficacy)	Stage IIIB or IV NSCLC	Phase II
NCT04735069	Binimetinib (Selective MEK inhibitor), Hydroxychloroquine (Lysosomotropic agent for autophagy blockage)	Binimetinib and Hydroxychloroquine in Patients with Advanced KRAS-mutant Non-Small Cell Lung Cancer	Recruiting	NSCLC, KRAS Mutation-Related Tumors	Phase II
NCT04620330	Avutometinib (VS-6766 a dual RAF/MEK inhibitor), Defactinib (FAK inhibitor)	A Study of Avutometinib (VS-6766) + Defactinib in Recurrent KRAS G12V, Other KRAS and BRAF Non-Small Cell Lung Cancer (RAMP202)	Recruiting	NSCLC, KRAS activating mutations	Phase II
NCT05074810	Avutometinib (VS-6766) and sotorasib (Direct KRAS inhibitor)	Phase 1/2 Study of Avutometinib (VS-6766) + Sotorasib in G12C NSCLC Patients (RAMP203)	Recruiting	NSCLC, KRAS activating mutations	Phase I/II
NCT05375994	Avutometinib (VS-6766) and Adagrasib (Direct KRAS inhibitor)	Study of Avutometinib (VS-6766) + Adagrasib in KRAS G12C NSCLC Patients (RAMP204)	Recruiting	NSCLC, KRAS activating mutations	Phase I/II
NCT02450656	Afatinib (ErbB inhibitor), Selumetinib (MEK inhibitor), Docetaxel (Chemotherapy agent)	Afatinib and Selumetinib in Advanced KRAS-mutant and PIK3CA Wildtype Non-small Cell Lung Cancer (M14AFS)	Unknown	NSCLC, Pancreatic Neoplasms, Gastrointestinal Neoplasms, Colorectal Neoplasms	Phase I/II
NCT03581487	Durvalumab (PD-1 inhibitor), Selumetinib (MEK1/2 inhibitor), Tremelimumab (CTLA-4 inhibitor)	Durvalumab, Tremelimumab, and Selumetinib in Treating Participants with Recurrent or Stage IV Non-small Cell Lung Cancer	Recruiting	Recurrent NSCLC, Stage IV/A/B Lung Cancer AJCC v8	Phase I/II
NCT03170206	Binimetinib (MEK1/2 inhibitor), Palbociclib (CDK4/6 inhibitor)	Study of the CDK4/6 Inhibitor Palbociclib (PD-0332991) in Combination with the MEK Inhibitor Binimetinib (MEK162) for Patients with Advanced KRAS-mutant Non-Small Cell Lung Cancer	Recruiting	Lung cancer	Phase I/II
NCT04967079	Trametinib (MEK1/2 inhibitor), Anlotinib (multi-targeted receptor tyrosine kinase inhibitor)	MEK Inhibitor Combined with Anlotinib in the Treatment of KRAS-mutated Advanced Non-small Cell Lung Cancer	Recruiting	NSCLC	Phase I
NCT02079740	Trametinib (MEK1/2 inhibitor), Navitoclax (Bcl-2 inhibitor)	Trametinib and Navitoclax in Treating Patients with Advanced or Metastatic Solid Tumors	Active, not recruiting	Metastatic/Refractory/Unresectable Malignant Solid Neoplasm	Phase I/II
NCT01248247	Erlotinib (EGFR), Selumetinib (MEK1/2 inhibitor), MK-2206 (inhibitor of pan-Akt), Sorafenib (inhibitor of RAF kinases, PDGFR, VEGFR)	BATTLE-2 Program: A Biomarker-Integrated Targeted Therapy Study in Previously Treated Patients with Advanced Non-Small Cell Lung Cancer	Completed	NSCLC	Phase II
NCT01951690	Defactinib (FAK inhibitor)	Phase II Study of VS-6063 in Patients with KRAS-mutant Non-Small Cell Lung Cancer	Completed	NSCLC	Phase II
NCT01124864	Luminespib (HSP90 inhibitor)	A Study of AUY922 in Non-small-cell Lung Cancer Patients Who Have Received Previous Two Lines of Chemotherapy	Completed	NSCLC	Phase II
NCT02142738	Pembrolizumab (PD-1 inhibitor), Paclitaxel (antimicrotubular agent), Carboplatin (Chemotherapy agent), Pemetrexed (Chemotherapy agent), Cisplatin (Chemotherapy agent), Gemcitabine (Chemotherapy agent)	Study of Pembrolizumab (MK-3475) Compared to Platinum-Based Chemotherapies in Participants with Metastatic Non-Small Cell Lung Cancer (MK-3475-024/KEYNOTE-024)	Completed	NSCLC	Phase III
NCT02775435	Pembrolizumab (PD-1 inhibitor), Paclitaxel (antimicrotubular agent), Carboplatin (Chemotherapy agent), Nab-Paclitaxel (Chemotherapy agent+ Albumin)	A Study of Carboplatin-Paclitaxel/Nab-Paclitaxel Chemotherapy with or Without Pembrolizumab (MK-3475) in Adults with First Line Metastatic Squamous Non-small Cell Lung Cancer (MK-3475-407/KEYNOTE-407)	Active, not recruiting	NSCLC	Phase III
NCT02367781	Atezolizumab (PD-1 inhibitor), Carboplatin (Chemotherapy agent), Pemetrexed (Chemotherapy agent), Nab-Paclitaxel (Chemotherapy agent+ Albumin)	A Study of Atezolizumab in Combination with Carboplatin Plus (+) Nab-Paclitaxel Compared with Carboplatin+Nab-Paclitaxel in Participants with Stage IV Non-Squamous Non-Small Cell Lung Cancer (NSCLC) (IMpower130)	Completed	NSCLC	Phase III
NCT02409342	Atezolizumab (PD-1 inhibitor), Carboplatin (Chemotherapy agent), Pemetrexed (Chemotherapy agent), Cisplatin (Chemotherapy agent), Gemcitabine (Chemotherapy agent)	A Study of Atezolizumab (MPDL3280A) Compared with a Platinum Agent (Cisplatin or Carboplatin) + (Pemetrexed or Gemcitabine) in Participants with Stage IV Non-Squamous or Squamous Non-Small Cell Lung Cancer (NSCLC) [IMpower110]	Completed	NSCLC	Phase III
NCT02366143	Atezolizumab (PD-1 inhibitor), Bevacizumab (VEGFR inhibitor), Carboplatin (Chemotherapy agent), Paclitaxel (antimicrotubular agent)	A Study of Atezolizumab in Combination with Carboplatin Plus (+) Paclitaxel with or Without Bevacizumab Compared with Carboplatin+Paclitaxel+Bevacizumab in Participants with Stage IV Non-Squamous Non-Small Cell Lung Cancer (NSCLC) (IMpower150)	Completed	NSCLC	Phase III
NCT02477826	Nivolumab (PD-1 inhibitor), Ipilimumab (CTLA-4 inhibitor), Paclitaxel (antimicrotubular agent), Carboplatin (Chemotherapy agent), Pemetrexed (Chemotherapy agent), Cisplatin (Chemotherapy agent), Gemcitabine (Chemotherapy agent)	An Investigational Immuno-therapy Trial of Nivolumab, or Nivolumab Plus Ipilimumab, or Nivolumab Plus Platinum-doublet Chemotherapy, Compared to Platinum Doublet Chemotherapy in Patients with Stage IV Non-Small Cell Lung Cancer (NSCLC) (CheckMate 227)	Active, not recruiting	NSCLC	Phase III
NCT03215706	Nivolumab (PD-1 inhibitor), Ipilimumab (CTLA-4 inhibitor), Paclitaxel (antimicrotubular agent), Carboplatin (Chemotherapy agent), Pemetrexed (Chemotherapy agent), Cisplatin (Chemotherapy agent)	A Study of Nivolumab and Ipilimumab Combined with Chemotherapy Compared to Chemotherapy Alone in First Line NSCLC (CheckMate 9LA)	Active, not recruiting	NSCLC	Phase III
NCT02125461	Durvalumab (PD-1 inhibitor)	A Global Study to Assess the Effects of MEDI4736 Following Concurrent Chemoradiation in Patients with Stage III Unresectable Non-Small Cell Lung Cancer (PACIFIC)	Active, not recruiting	NSCLC	Phase III
NCT03164616	Durvalumab (PD-1 inhibitor), Tremelimumab (CTLA-4 inhibitor), Abraxane + carboplatin, Gemcitabine + cisplatin, Gemcitabine + carboplatin, Pemetrexed + carboplatin, Pemetrexed + cisplatin	Study of Durvalumab + Tremelimumab with Chemotherapy or Durvalumab with Chemotherapy or Chemotherapy Alone for Patients with Lung Cancer (POSEIDON). (POSEIDON)	Active, not recruiting	NSCLC	Phase III
NCT03447769	Canakinumab (Anti IL-1β)	Brief Title: Study of Efficacy and Safety of Canakinumab as Adjuvant Therapy in Adult Subjects with Stages AJCC/UICC v. 8 II-IIIA and IIIB (T > 5 cm N2) Completely Resected Non-small Cell Lung Cancer Acronym: CANOPY-A (Canopy-A)	Active, not recruiting	NSCLC	Phase III
NCT03968419	Canakinumab (Anti IL-1β), Pembrolizumab (PD-1 inhibitor)	This Study Will Evaluate the Effect of Canakinumab or Pembrolizumab Given as Monotherapy or in Combination as Neo-adjuvant Treatment for Subjects with Early Stages NSCLC. (CANOPY-N)	Completed	NSCLC	Phase II
NCT04905316	Canakinumab (Anti IL-1β), Durvalumab (PD-1 inhibitor), Radiation therapy, Chemotherapy	A Study of Canakinumab with Chemotherapy, Radiation Therapy, and Durvalumab in People with Lung Cancer (CHORUS)	Recruiting	NSCLC	Phase II
NCT03626545	Canakinumab (Anti IL-1β), Docetaxel (antimicrotubular agent)	Phase III Study Evaluating Efficacy and Safety of Canakinumab in Combination with Docetaxel in Adult Subjects with Non-small Cell Lung Cancers as a Second- or Third-Line Therapy (CANOPY-2)	Completed	NSCLC	Phase III
NCT04691817	Atezolizumab (PD-1 inhibitor), Tocilizumab (IL-6 inhibitor)	Tocilizumab and Atezolizumab in Adults with Locally Advanced or Metastatic Non-Small Cell Lung Cancer Refractory to 1st Line Immune Checkpoint Inhibitor-Based Therapy	Not yet recruiting	NSCLC	Phase I/II
NCT04940299	Atezolizumab (PD-1 inhibitor), Ipilimumab (CTLA-4 inhibitor), Nivolumab (PD-1 inhibitor)	Tocilizumab, Ipilimumab, and Nivolumab for the Treatment of Advanced Melanoma, Non-Small Cell Lung Cancer, or Urothelial Carcinoma	Recruiting	NSCLC, Melanoma, Urothelial Cancer	Phase II
NCT03337698	Atezolizumab (PD-1 inhibitor), Tocilizumab (IL-6 inhibitor), Carboplatin (Chemotherapy agent), Pemetrexed (Chemotherapy agent), Docetaxel (antimicrotubular agent), Gemcitabine (Chemotherapy agent), Cobimetinib (MEK1/2 inhibitor), RO6958688 (Anti CEA-CD3), CPI-444 (Adenosine A2A receptor inhibitor), Linagliptin (dipeptidyl peptidase-4 (DPP-4) inhibitors), Ipatasertib (AKT inhibitor), Bevacizumab (VEGFR inhibitor), Sacituzumab Govitecan (Topoisomerase inhibitor), Evolocumab (Proprotein convertase subtilisin kexin type 9 inhibitor), Radiation	A Study of Multiple Immunotherapy-Based Treatment Combinations in Participants with Metastatic Non-Small Cell Lung Cancer (Morpheus- Non-Small Cell Lung Cancer) (Morpheus Lung)	Recruiting	NSCLC	Phase I/II
NCT04879849	Pembrolizumab (PD-1 inhibitor), TAK-676 (STING agonist), Image-guided radiation therapy	A Study of TAK-676 With Pembrolizumab After Radiation Therapy to Treat a Number of Cancers	Recruiting	NSCLC, Triple Negative Breast Neoplasms, Squamous Cell Carcinoma of Head and Neck	Phase I
NCT01184807	OPB-51602 (STAT3 inhibitor)	Phase 1, Dose-escalation Trial of OPB-51602 in Patients with Advanced Solid Tumors	Completed	Malignant solid tumor	Phase I
NCT03195699	TTI-101 (STAT3 inhibitor)	Oral STAT3 Inhibitor, TTI-101, in Patients with Advanced Cancers	Active, not recruiting	NSCLC, Colorectal Cancer, Gastric Adenocarcinoma, Melanoma, Hepatocellular Cancer, Head and Neck Squamous Cell Carcinoma, Breast Cancer	Phase I
NCT02983578	Danvatirsen (Antisense oligonucleotide inhibitor of STAT3), Durvalumab (PD-1 inhibitor)	Danvatirsen and Durvalumab in Treating Patients with Advanced and Refractory Pancreatic, Non-Small Cell Lung Cancer, and Mismatch Repair Deficient Colorectal Cancer	Active, not recruiting	NSCLC, Advanced Colorectal Carcinoma, Mismatch Repair Deficiency, Refractory and Advanced Pancreatic Carcinoma	Phase II

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
