# Peer review of "KRAS-Mutant Lung Cancer: Targeting Molecular and Immunologic Pathways, Therapeutic Advantages and Restrictions"

_cells, 2023, doi:10.3390/cells12050749_

Round 1

Reviewer 1 Report

I apologize for the delay in submitting your review.

This review article summarizes the evidence for treatment of the NSCLC patients with K-ras mutations.

It contains a great deal of information and is systematically written, so I think it will be a very useful review article for the reader.

Although there are no major corrections to this article, it would be better to have a table with information about the clinical trials described in this article.

Author Response

Comments:

This review article summarizes the evidence for treatment of the NSCLC patients with K-ras mutations.

It contains a great deal of information and is systematically written, so I think it will be a very useful review article for the reader.

Although there are no major corrections to this article, it would be better to have a table with information about the clinical trials described in this article.

Responses:

Thank you for your comments and great suggestion of having a table summarizing the clinical trials. We agree with this suggestion, therefore we have provided Tables 1, and 2 in the revised version.

Reviewer 2 Report

The aim of this review is to provide an overview of the success and pitfalls of the various therapeutics targeting KRAS mutant lung cancer that are currently under investigation in preclinical and clinical settings. There have been a lot of review articles published in the last three years covering the same topic; while the authors have tried to summarize much of the current KRAS molecular targets being investigated, the review lacks some cohesion and insight on how this information could be used to design novel agents/combinations for this disease, and needs to be revised as follows:

1.       The authors could consider expanding section 2.2 to include KRAS mutations/co-mutations reported between genders, ethnicities as well as patients with family or prior history of respiratory conditions.

2.       Section 3 may need some restructuring to improve cohesion and flow. Consider having major headings in following order: Direct KRAS targeting, Indirect KRAS targeting (including Upstream targets (EGFR), KRAS membrane Association, Downstream targets), Targeted Metabolic reprograming (Metabolic rewriting and autophagy), Synthetic lethality partners inhibitors) and Immunotherapies.

3.       Direct KRAS inhibitors should include some information other KRASG12C allosteric inhibitors like D-1553 (NCT04585035), LY3537982(NCT04956640), GDC-6036 (NCT04449874), JDQ443 (NCT04699188), JAB-21822 (NCT05002270), and LY3499446 and ARS-3248/JNJ-74699157 (NCT04006301 and NCT04585035). As well as some information about Ras/Rap1-specific endopeptidase (RRSP) (PMID: 30279169, 32616570).

4.       The authors could consider including a separate section describing the mechanisms of acquired on-target and off-target resistance to KRAS inhibitors identified to date or include this part in each of the subheadings of section 3.

5.       In addition to detailing the challenges of the therapeutics described in section 3, the authors should also highlight the various combination treatments being tested between the inhibitors described. For example: Combination of MEK with various inhibitors (PMID: 35658005, 35658005, 35658005, NCT02079740), HSP90 and autophagy (PMID: 35658005), Direct KRAS inhibitors and immunotherapies/other inhibitors (PMID: 31666701, 35811500, 35872166, NCT04185883, NCT03785249, NCT04111458, NCT01229150, NCT02230553). This information could be presented in a Table

6.       Section 4 and discussion only discusses co-mutations and immunotherapies – this needs to be expanded to include other inhibitors discussed as well. The authors could also consider merging section 4 into section 3.

7.       Lines 92-101 could be expanded to include the following articles: PMID: 29089357 and 34684076

8.       While Figure 2 summarizes all the inhibitors discussed in the review, it appears like these inhibitors function independent of one another – providing a pathway diagram demonstrating the various targets of these inhibitors may be more informative here for the reader.

9.       In-text citations for both figures are missing and some proof-reading is required.

Author Response

Comment 1: The authors could consider expanding section 2.2 to include KRAS mutations/co-mutations reported between genders, ethnicities as well as patients with family or prior history of respiratory conditions

Response: Thank you for pointing this out. We agree with this comment. Therefore, we have added the required information as suggested in the revised manuscript, section 2.2 line 108-126.

Comment 2: Section 3 may need some restructuring to improve cohesion and flow. Consider having major headings in following order: Direct KRAS targeting, Indirect KRAS targeting (including Upstream targets (EGFR), KRAS membrane Association, Downstream targets), Targeted Metabolic reprograming (Metabolic rewriting and autophagy), Synthetic lethality partners inhibitors) and Immunotherapies.

Response: We agree with this suggestion and have incorporated it throughout the revised manuscript.

Comment 3: Direct KRAS inhibitors should include some information other KRASG12C allosteric inhibitors like D-1553 (NCT04585035), LY3537982(NCT04956640), GDC-6036 (NCT04449874), JDQ443 (NCT04699188), JAB-21822 (NCT05002270), and LY3499446 and ARS-3248/JNJ-74699157 (NCT04006301 and NCT04585035). As well as some information about Ras/Rap1-specific endopeptidase (RRSP) (PMID: 30279169, 32616570).

Response: As instructed, we have accordingly, prepared table 1 to include other KRASG12C inhibitors and added some information about RRSP in section 3.1. line 190-198 to emphasize this point.

Comment 4: The authors could consider including a separate section describing the mechanisms of acquired on-target and off-target resistance to KRAS inhibitors identified to date or include this part in each of the subheadings of section 3.

Response: We agree with this suggestion and have incorporated it throughout the manuscript providing a new section as section 4 with the mechanisms of acquired on-target and off-target resistance to KRAS inhibitors line 565-594.

Comment 5: In addition to detailing the challenges of the therapeutics described in section 3, the authors should also highlight the various combination treatments being tested between the inhibitors described. For example: Combination of MEK with various inhibitors (PMID: 35658005, 35658005, 35658005, NCT02079740), HSP90 and autophagy (PMID: 35658005), Direct KRAS inhibitors and immunotherapies/other inhibitors (PMID: 31666701, 35811500, 35872166, NCT04185883, NCT03785249, NCT04111458, NCT01229150, NCT02230553(colorectal cancer)). This information could be presented in a Table.

Response: As instructed, we now provided table 2 to summarize all the clinical trials mentioned in the text in addition to those suggested by the reviewer such as combinational therapy. However, we did not mention NCT2230553 as it was an evaluation in Colorectal cancer patients.

Comment 6: Section 4 and discussion only discusses co-mutations and immunotherapies – this needs to be expanded to include other inhibitors discussed as well. The authors could also consider merging section 4 into section 3.

Response: Thanks for the suggestion. We have now included some of the suggested information as instructed in section 5 line 598-603.

Comment 7: Lines 92-101 could be expanded to include the following articles: PMID: 29089357 and 34684076

Response: Both articles now included in section 2.2 line 100-111.

Comment 8: While Figure 2 summarizes all the inhibitors discussed in the review, it appears like these inhibitors function independent of one another – providing a pathway diagram demonstrating the various targets of these inhibitors may be more informative here for the reader.

Response: You have raised an important point here. However, we believe that Figure 1 could provide the information related to pathways which are targets of some of the inhibitors.

Comment 9: In-text citations for both figures are missing and some proof-reading is required

Response: Thank you for pointing this out. We have accordingly provided the missing in text citations and proof-reading in both figures.

Reviewer 3 Report

The authors of this article review recent studies addressed to find therapeutic options for Non-Small Cell Lung Cancer (NSCLC) patients that present KRAS mutations in the tumour. The review is very extensive and consider molecules inhibiting KRAS directly, pathways in which KRAS is involved such as metabolism and autophagy and also downstream KRAS effectors (RAF. MEK, PI3K, AKT, mTOR, FAK and HSP90). In addition, immunotherapies are also reviewed. Finally, the authors discuss the role of co-mutations and their impact in therapeutic outcomes.

The review article is very well written and supported by 125 references which will be very useful for the researchers working in this interesting field of NSCLC personalized treatment. There are just a few comments that could help to improve the review, as follows.

Major points.

1. In the Immunotherapies section I have missed some reference to the possible use of Chimeric Antigen Receptor T cells (CARTs). This therapy is starting to be extensively used in some tumors and I think that providing some considerations or discussion about its possible use in NSCLC would be very useful to readers of the review.

2. The review is very extensive and contains a lot of information. Because of this reasons it could be useful to summarize the more clinically relevant information in one table.

Minor points:

1. Abstract, line 22. “under investigations” could be changed to “under investigation”

2. Line 63 “GTP-GDP substation” might be “GTP-GDP substitution”

3. Line 172, “RNA interferes” might be “RNA interferent” or “RNA interference”

4. Lines 209-211. This sentence is difficult to understand and should be revised.

5. Line 406 “IL-1RI and IL-2RII” probably should be “IL-1RI and IL-1RII”

Author Response

Major points:

Comment 1: In the Immunotherapies section I have missed some reference to the possible use of Chimeric Antigen Receptor T cells (CARTs). This therapy is starting to be extensively used in some tumors and I think that providing some considerations or discussion about its possible use in NSCLC would be very useful to readers of the review

Response: Thank you for this suggestion. We agree that CAR-T cell therapy is an important approach in treatment of cancer which is gaining a lot of attention these days. However, we found that slightly hard to find enough resources to include CAR-T cells therapy in Kras mutant NSCLC, which is the focus of our review article. Hence, we have limited our review of this approach to a small section (3.6.3 line 541-555) in our revised review.

Comment 2: The review is very extensive and contains a lot of information. Because of this reasons it could be useful to summarize the more clinically relevant information in one table.

Response: Thank you for pointing this out. We agree with this comment. Therefore, we have provided 2 tables, including direct KRAS inhibitor in table 1 and other therapies in table 2 with the revised version.

Minor points:

Comment 1: Abstract, line 22. “under investigations” could be changed to “under investigation”

Response:  Thank you for pointing this out. We corrected investigations with investigation in line 22.

Comment 2: Line 63 “GTP-GDP substation” might be “GTP-GDP substitution” 

Response: Thank you for your attention. We changed substation with substitution in line 63.

Comment 3: Line 172, “RNA interferes” might be “RNA interferent” or “RNA interference”

Response: Agree. We changed RNA interferes with RNA interferent which is in line 346 of revised manuscript.

Comment 4: Lines 209-211. This sentence is difficult to understand and should be revised. 

Response: We agree and have rephrased the sentence for more clarity. Line 211-213 of revised manuscript.

Comment 5: Line 406 “IL-1RI and IL-2RII” probably should be “IL-1RI and IL-1RII”

Response: Agreed and changed to IL-IRII in line 455 of revised manuscript.

Round 2

Reviewer 2 Report

The authors have adequately modified the manuscript based on the reviewer feedback. Very minor comments remaining:

1. The authors should consider including the inhibitors (or atleast the top 10) in the KRAS pathway. Although Figure 2 summarizes all the inhibitors, having a visual image of the targets of some of these inhibitors would be very useful and increase the impact of this manuscript. 

2. There are some typos still remaining in the revised manuscript (For example" Line 346 - RNA interferent should be changed to RNA interference)

Author Response

The authors have adequately modified the manuscript based on the reviewer feedback. Very minor comments remaining:

Comment 1: The authors should consider including the inhibitors (or atleast the top 10) in the KRAS pathway. Although Figure 2 summarizes all the inhibitors, having a visual image of the targets of some of these inhibitors would be very useful and increase the impact of this manuscript. 

Response: Thank you for your feedback, we agree to this comment therefore we included inhibitors into figure 1 (KRAS pathways).

Comment 2: There are some typos still remaining in the revised manuscript (For example" Line 346 - RNA interferent should be changed to RNA interference)

Response:  We agree therefore we revised the manuscript for typos and you suggested.

Line 346- RNA interferent changed to RNA interference

Line 383- defence changed to defense.

Line 575- habour changed to habor.